# Long-Range PCR-Based NGS Applications to Diagnose Mendelian Retinal Diseases

**DOI:** 10.3390/ijms22041508

**Published:** 2021-02-03

**Authors:** Jordi Maggi, Samuel Koller, Luzy Bähr, Silke Feil, Fatma Kivrak Pfiffner, James V. M. Hanson, Alessandro Maspoli, Christina Gerth-Kahlert, Wolfgang Berger

**Affiliations:** 1Institute of Medical Molecular Genetics, University of Zurich, 8952 Schlieren, Switzerland; maggi@medmolgen.uzh.ch (J.M.); koller@medmolgen.uzh.ch (S.K.); baehr@medmolgen.uzh.ch (L.B.); feil@medmolgen.uzh.ch (S.F.); kivrakpfiffner@medmolgen.uzh.ch (F.K.P.); maspoli@medmolgen.uzh.ch (A.M.); 2Department of Ophthalmology, University Hospital Zurich and University of Zurich, 8091 Zurich, Switzerland; james.hanson@usz.ch (J.V.M.H.); christina.gerth-kahlert@usz.ch (C.G.-K.); 3Zurich Center for Integrative Human Physiology (ZIHP), University of Zurich, 8057 Zurich, Switzerland; 4Neuroscience Center Zurich (ZNZ), University and ETH Zurich, 8057 Zurich, Switzerland

**Keywords:** genetic testing, retinal diseases, sequencing, NGS, diagnostics, long-range PCR, CNV, phasing, missing heritability, *ABCA4*, *PRPH2*, *BEST1*

## Abstract

The purpose of this study was to develop a flexible, cost-efficient, next-generation sequencing (NGS) protocol for genetic testing. Long-range polymerase chain reaction (PCR) amplicons of up to 20 kb in size were designed to amplify entire genomic regions for a panel (*n* = 35) of inherited retinal disease (IRD)-associated loci. Amplicons were pooled and sequenced by NGS. The analysis was applied to 227 probands diagnosed with IRD: (A) 108 previously molecularly diagnosed, (B) 94 without previous genetic testing, and (C) 25 undiagnosed after whole-exome sequencing (WES). The method was validated with 100% sensitivity on cohort A. Long-range PCR-based sequencing revealed likely causative variant(s) in 51% and 24% of proband from cohorts B and C, respectively. Breakpoints of 3 copy number variants (CNVs) could be characterized. Long-range PCR libraries spike-in extended coverage of WES. Read phasing confirmed compound heterozygosity in 5 probands. The proposed sequencing protocol provided deep coverage of the entire gene, including intronic and promoter regions. Our method can be used (i) as a first-tier assay to reduce genetic testing costs, (ii) to elucidate missing heritability cases, (iii) to characterize breakpoints of CNVs at nucleotide resolution, (iv) to extend WES data to non-coding regions by spiking-in long-range PCR libraries, and (v) to help with phasing of candidate variants.

## 1. Introduction

Inherited retinal diseases (IRDs) are a group of disorders affecting the retina and its function. They are characterized by high phenotypic variability and genetic heterogeneity, including different modes of inheritance [1]. These disorders may present as either isolated or syndromic, progressive or stationary. Also, IRDs may affect the entire retina (panretinal) or may be restricted to the macula. Moreover, they are often classified based on the primarily affected photoreceptor type. Panretinal forms include retinitis pigmentosa (RP), cone-rod dystrophy (CRD), cone dystrophy (COD), and others, whereas macular dystrophies (MDs) include Stargardt disease (STGD), Best disease (BEST), and others [1,2].

Due to the multitude of loci associated with IRDs (almost 300 loci, https://sph.uth.edu/RetNet), molecular diagnostics often relies on targeted enrichment and high-throughput sequencing, either whole-exome (WES) [3,4,5,6] or gene panels [5,6,7,8,9,10]. Studies have reported that diagnostic yield when using these standard methods (WES/gene panels) is typically between 50% and 76%, depending, amongst other factors, on the specific clinical subtype being investigated [6,7,8,9,11,12,13,14].

Missing heritability is defined as the portion of the phenotypic trait that could not (yet) be explained by genotype data [15]. Since inherited diseases are, by definition, characterized by complete or near-complete heritability, missing heritability in these diseases indicates cases in which genetic testing could not identify a molecular diagnosis [15].

Recent efforts to reduce missing heritability in IRDs (and improve diagnostic output) have focused on non-coding regions of the genome, and on the detection of structural or copy number variants (CNVs) [16,17,18,19]. These studies were performed using customized microarray probes [17], customized capturing probes [16,18], or genome sequencing [19].

Several studies have focused on the *ABCA4* locus [16,18,20,21,22,23], as it has often been reported to be one of the most prevalent mutated genes in IRD cohorts [10,12,13,14,24]. Variants in this gene are the cause of STGD1 (MIM 248200) [25]. Previous studies have reported that 15–40% of STGD patients remain without a molecular diagnosis after standard genetic testing [16,18,20,21,22,23,26]. These studies identified CNVs and several non-canonical splice site and deep-intronic variants that have been shown to affect splicing, findings that partially explain missing heritability in STGD [16,18,20,21,22,23].

Here, we present a flexible and cost-effective method to comprehensively sequence loci of interest. The method relies on high-throughput sequencing of pooled long-range (LR) polymerase chain reaction (PCR) fragments of up to 20 kb in size. We designed primers to sequence 35 genomic loci that have been associated with IRDs, particularly MDs and X-linked RP, including *ABCA4*, *PRPH2*, *BEST1*, *CRB1*, *CNGB3*, *RP1*, and *RPGR.*

## 2. Results

LR PCRs were established for 35 loci associated with IRDs. The total genomic target equals 1.81 Mb, and the longest locus is *CRB1*, with a total genomic target of 212 kb (Table 1). The entire regions of most target genes could be amplified and sequenced. Several intronic regions could not be amplified after several attempts: *CRB1* (3.8 kb in size, NC_000001.10:g.197420721–197424549), *EFEMP1* (0.5 kb, NC_000002.11:56102261–56102747), *IMPG1* (1.9 kb, NC_000006.11:76729633–76731567 and 1.6 kb, NC_000006.11:76633571–76635199), *CNGB3* (7.1 kb, NC_000008.10:g.87647022–87654129 and 3.8 kb, NC_000008.10:87625835–87629598), and *TIMP3* (1.5 kb, NC_000022.10:33202359–33203885).

The primers for loci that require multiple PCRs were designed in a way that amplicons of adjacent regions overlap with each other. As an example, Figure 1 shows the coverage obtained for the *ABCA4* locus by different NGS assays: WES, custom capture probes, and LR PCRs. 

As per design, WES only provides coverage for the coding portions of the locus. The custom capture probes result in coverage for the majority of the locus, including introns. However, gaps accounting for 15–20% of the locus are present. Finally, the LR PCR method can provide uniform coverage over the entire locus. Overlaps between neighboring LR amplicons are revealed by an increase in coverage. Similar results were obtained for the other loci included in the panel (Appendix A in the Appendix A).

In order to verify sensitivity of the method, 108 IRD patients were selected, for whom a molecular diagnosis had been previously established, corresponding to variants in a locus included in the panel. LR PCRs for the respective loci were performed and sequenced. The sequencing of the validation cohort resulted in 100% sensitivity to the previously identified putative pathogenic variants (Table A1 in the Appendix B). We found no evidence for allele dropout (ADO). 

We envisioned the method to be useful in the following scenarios.

### 2.1. First-Tier Assay for Probands without Previous Genetic Testing

A subpanel of the 35 loci was selected and sequenced for 94 non-syndromic IRD cases composed of probands diagnosed with MD (92/94), CRD (1/94), and autosomal dominant COD (1/94) that had not previously undergone genetic testing. Locus selection was based on the provided clinical diagnosis and family history. This strategy resulted in the identification of a molecular diagnosis in 48/94 probands (51.1%) (Table A2 in the Appendix B).

If candidate loci sequencing did not reveal a molecular diagnosis, WES was performed. Second-tier WES analysis revealed likely pathogenic variants in 15 additional probands, resulting in a total of 63/94 molecularly diagnosed probands (Table A2 in the Appendix B). Thirty-one probands (33.0%) were still lacking a genetic diagnosis after first-tier LR PCR-based candidate loci sequencing and second-tier WES. A list of rare variants (gnomAD MAF < 1%) found in these samples is provided (Appendix A in the Appendix A).

### 2.2. Tackling Missing Heritability

The missing heritability cohort included 25 probands, in whom WES was previously performed and a single likely pathogenic variant in a recessively inherited locus was identified. LR PCRs of the relevant loci (i.e., those carrying a single likely pathogenic variant) were performed to assess the presence of a second likely pathogenic variant.

This strategy led to the discovery of additional likely pathogenic variants in *ABCA4* in 6/25 probands (23.1%). The most common likely causative variant identified is NM_000350.2:c.5603A > T (3 alleles), which is often excluded from analysis due to the relatively high minor allele frequency in gnomAD (6.65% in Non-Finnish European) [18]. The remaining cases could be explained by the discovery of previously published deep intronic variants: NM_000350.2:c.4253 + 43G > A [21,22], NM_000350.2:c.4539 + 2064C > T [18,20], and NM_000350.2:c.5196 + 1056A > G [16,20] (Table A3 in the Appendix B). Moreover, this strategy led to the detection of several deep-intronic variants of unknown significance, which require further analyses for interpretation of pathogenicity (Appendix A in the Appendix A).

### 2.3. Copy Number Variant (CNV) Characterization

CNV analysis on capture data allowed for the identification of likely pathogenic exon-spanning deletions in *ABCA4*, *KCNV2*, *RP1*, and *RS1*.

A heterozygous *ABCA4* deletion spanning exons 20 through 22 and parts of the adjacent introns was identified in two unrelated STGD patients and confirmed by multiplex ligation-dependent probe amplification (MLPA; Figure A1 and Table A2). A LR PCR product generated by using forward and reverse primers upstream of exon 19 and downstream of exon 23, respectively, was sequenced on a MiSeq instrument (Illumina, San Diego, CA, USA). Results highlighted similar breakpoints as in a recently published recurrent deletion [18]; however, mismatches and a longer polyadenine stretch seem to be present around the breakpoints. The breakpoint region was also sequenced by targeted locus amplification (TLA; Cergentis, Utrecht, The Netherlands), which appeared to confirm the 3’ breakpoint at NC_000001.10:g.94511701. The 5’ breakpoint probably lies within an A-rich sequence at NC_000001.10:g.94507655–94507699. Interestingly, the breakpoints are flanked by an AluY and an AluSx element on the 5′- (NC_000001.10:g.94507370–94507684) and 3′-side (NC_000001.10:g.94511717–94512041), respectively. Finally, a 10 bp microhomology sequence directly follows the 5′-side AluY element (NC_000001.10:g.94507690–94507699) and precedes the 3′-side AluSx element (NC_000001.10:g.94511701–94511710). However, unambiguous breakpoint identification was not possible due to repetitive flanking sequences (Table A2 and Figure A1).

A heterozygous deletion spanning the entire *KCNV2* locus was identified in a patient diagnosed with COD (Table A1 in the Appendix B). To narrow down the breakpoint region of the deletion, microarray analysis was performed with an 850K single nucleotide polymorphism (SNP) chip. Based on these results, primers were designed for LR and the resulting amplicon was sequenced. Sequence alignment revealed a deletion of 70,036 bp in size and a 3 bp microhomology sequence flanking the breakpoints (Table A1 and Figure A2).

Finally, a hemizygous deletion involving exon 2 of *RS1* and a homozygous deletion spanning exons 2–4 of *RP1* were identified in a X-linked retinoschisis (XLRS) patient and a RP patient, respectively. The breakpoints of these deletions could be determined directly with the validated LR PCRs (Appendix A in the Appendix A). The *RP1* deletion is 11,116 bp in length with a 3 bp microhomology region flanking the breakpoints and has been described previously [27] (Table A1 in the Appendix B). The *RS1* deletion is 1005 bp in length and features a 4 bp microhomology around the breakpoints (Table A1 and Figure A3).

### 2.4. Exome Spike-In

In order to include the intronic regions of *ABCA4*, custom-capture probes were added to the TruSeq Exome kit capture probes before the hybridization step. The resulting data provided low coverage of the intronic regions of the gene (Appendix A in the Appendix A). However, achieving the appropriate dilutions and proportions to ensure that the *ABCA4* capture probes do not outnumber WES probes was challenging.

Similarly, LR PCR libraries spike-in was performed for the *ABCA4*, *RPGR*, and *PCARE* (*C2orf71*), to either increase coverage of poorly captured regions, such as *RPGR*’s exon ORF15 (open reading frame 15) and *PCARE* exon 1, or to cover non-captured regions, such as introns. The method provided reliable data for variant detection over the entire loci, including exon ORF15 (Appendix A in the Appendix A). Moreover, it proved to be technically less challenging compared to custom probes spike-in. Finally, the extra target regions can be personalized for each individual sample.

### 2.5. Read Phasing

In case of compound heterozygosity for recessive variants, chromosomal phasing for the candidate variants should be performed. Segregation analysis can achieve this, but only if family members are available for testing. Additional family members of 66 probands were available for segregation analysis in this study (66/227, 29.1%).

Being able to perform chromosome phasing directly on an individual’s sample would greatly help with interpretation of genetic testing results. By visually inspecting reads resulting from LR PCR sequencing (read phasing), it was possible to ascertain compound heterozygosity in 5 cases (Table A1, Table A2 and Table A3 in the Appendix B). Figure 2 shows a graphic representation of the concept (a), along with the simplest specific example (patient ID S220, Table A2) (b).

### 2.6. Macular Dystrophies (MD) Cohort

MD cases were present in all three groups analyzed in this study: 70 in the validation cohort, 13 in the missing heritability group, and 85 in the cohort in whom genetic testing had not been previously performed. The MD cohort (N = 168) was composed of probands with clinical diagnosis of STGD (*n* = 95), unspecified MD (*n* = 45), Best disease (*n* = 11), malattia leventinese (ML; *n* = 9), XLRS (*n* = 6), Sorsby fundus dystrophy (SFD; *n* = 1), and adult vitelliform macular dystrophy (AVMD; *n* = 1) (Table 2). The overall diagnostic yield equals 78.6%, with the STGD sub-cohort reaching 88.4% detection rate (Table 2 and Figure 3). Moreover, similar to previously reported findings [13,14], *ABCA4* (*n* = 75), *PRPH2* (*n* = 17), and *BEST1* (*n* = 10) are the most common contributors in our cohort (Figure 3).

A total of 161 unique, likely pathogenic variants were deemed disease-relevant in our cohort, with 39 being novel (not found in Human Gene Mutation Database; summarized in Table 3). Of these 161 variants, 127 were single nucleotide variants (26 novel), 30 small indels (10 novel), and 4 larger deletions (3 novel). The most common causal variants were NM_000350.2(*ABCA4*):c.5882G > A (35 alleles), NM_000322.4(*PRPH2*):c.514C > T (11 alleles), NM_000350.2(*ABCA4*):c.5603A > T (not in *cis* with other known causal variants, 10 alleles), NM_000350.2(*ABCA4*):c.5714 + 5G > A (9 alleles)*,* NM_000350.2(*ABCA4*):c.[5461–10T > C;5603A > T] (7 alleles), NM_000350.2(*ABCA4*):c.[1622T > C;3113C > T] (6 alleles), and NM_000350.2(*ABCA4*):c.[2588G > C;5603A > T] (6 alleles). Among the novel findings (Table 3), there are three recurrent likely pathogenic variants: NM_000350.2(*ABCA4*):c.4958G > A (2 alleles), NM_152778.2(*MFSD8*):c.670A > T (3 alleles), and NM_133497.3(*KCNV2*):c.1096del (2 alleles, found once in an undiagnosed case).

## 3. Discussion

We have presented a targeted sequencing approach based on LR PCR products that is flexible, versatile, and cost-effective. Primers for 124 LR PCRs to cover 35 IRD-associated loci have been established. The method has been validated with 108 molecularly diagnosed IRD cases. All previously identified variants could be verified, corresponding to 100% sensitivity. The target regions to be sequenced may be personalized according to the clinical phenotype and family history, so that costs can be reduced compared to standard genetic testing.

Although the method provides the most complete coverage of target loci (Figure 1, Appendix A in the Appendix A), several intronic regions could not be amplified (20 kb out of 1815 kb, 0.6%). Therefore, few loci (*CRB1*, *EFEMP1*, *IMPG1*, *CNGB3*, and *TIMP3*) have minor gaps in coverage. Moreover, whilst the method is very sensitive to deletions that lie within the amplicons, it is not able to detect deletions spanning primer-binding regions. In this case, ADO would occur, and the assay would not show any sign of the CNV other than lower PCR output and the fact that affected regions display exclusively homozygous-appearing variants.

It is not possible to eliminate the risk of ADO in any PCR; however, we did not observe any such events. Homozygous-appearing regions in non-consanguineous families should warrant caution. A final limitation of the method is the need for high molecular weight DNA as a template.

Since the sequencing targets are selected based on clinical phenotype and family history, this method is highly dependent on precise clinical assessment. Close collaboration between the molecular and the clinical diagnostics teams is therefore vital.

The method was used as a first-tier assay to sequence a subpanel of the 35 loci, selected based on clinical diagnosis, in 94 IRD patients (mostly MD) that did not undergo genetic testing previously. Additionally, LR PCR-based sequencing was used as a second-tier assay to discover “second hits” in 25 IRD patients of the missing heritability cohort. These strategies resulted in the identification of likely pathogenic variants in 69 patients of the 119 (58.0%). Published deep-intronic variants in *ABCA4* contributed to the diagnosis of 4 probands (3.4%).

Variants in *ABCA4*, *PRPH2*, and *BEST1* alone explained retinal disease in 60.7% of the MD cohort, similar to previously published results (57%) [13]. Even more striking are the results for the STGD sub-cohort, where variants in *ABCA4* and *PRPH2* were found to be the likely cause of disease in 76.8% of probands (Figure 3). For this reason, we envision our method to be particularly applicable to diseases with low genetic heterogeneity (such as STGD), and as part of a tiered genetic testing strategy to reduce the number of more costly assays (such as WES) for more genetically heterogeneous diseases (such as unspecified MD and Leber congenital amaurosis). In our study, use of a tiered strategy composed of LR PCR sequencing of *ABCA4*, *PRPH2*, and *BEST1*, followed by WES for the remaining undiagnosed samples, in the MD cohort would have saved 34% in material costs. A similar system was previously found to be more sensitive and less expensive than standard WES analysis in one of the largest IRD cohorts yet reported [12].

Furthermore, the protocol can be useful in analyzing challenging regions that are typically not covered by other methods, such as *RPGR*’s exon ORF15. It has been shown previously that NGS of a LR PCR over this region provides good coverage and we have developed a secondary data analysis pipeline to improve sensitivity and specificity [28,29]. This strategy permitted the identification of a novel 20 bp insertion (NM_001034853.1:c.2819_2838dup, Table 3, Table A1) that was not detected by standard analysis pipelines.

As demonstrated by the examples of the *ABCA4*, *RP1*, *RS1*, and *KCNV2* deletions, the protocol can be adapted easily to characterize the breakpoints of identified CNVs. Even though unambiguous breakpoint characterization was not possible for the *ABCA4* deletion, sequencing revealed the deletion to be flanked by two Alu elements and a 10 bp microhomology sequence. The two flanking Alu elements are characterized by variants that increase their homologies, which is suggestive of a gene conversion event [30]. Since the likelihood of these events is indirectly correlated with the distance of the elements, it may be secondary to the deletion event [30].

As a proof of concept, we showed that finalized LR libraries can be spiked into an exome library to either enhance coverage of poorly captured exonic regions (such as *RPGR*’s exon ORF15) or to obtain coverage of otherwise uncaptured regions (such as *ABCA4*’s introns) (Appendix A in the Appendix A).

Finally, the method can facilitate chromosomal phasing. It allows for segregation analysis of multiple variants on the same locus in a single experiment, when samples from family members are available. Moreover, when family members are not available for testing, read phasing based on the LR PCR sequencing data of the index patient might confirm or exclude compound heterozygosity. However, this depends on the density of heterozygous informative variants in the region of interest (Figure 2).

The method may also be beneficial for other Mendelian diseases, such as *CFTR*-related diseases, Tay-Sachs disease, Marfan syndrome, and tuberous sclerosis.

## 4. Materials and Methods

### 4.1. Patients and Family Members

Unrelated probands (N = 227) were referred to us for genetic testing from different eye clinics with a clinical diagnosis and information about family history. Samples (N = 166) from 66 families were available for segregation analysis. Written informed consent was obtained from all probands and family members included in this study, which was conducted in accordance with the 2013 Declaration of Helsinki.

The probands included in this study were assigned to one of three groups: (i) validation cohort (molecular diagnosis previously established by using WES, *n* = 108), (ii) missing heritability cohort (no molecular diagnosis established with previous WES analysis, *n* = 25), (iii) probands without previous genetic testing (*n* = 94).

### 4.2. Genomic DNA

The majority of genomic DNA (gDNA) samples (*n* = 204) were extracted in duplicate with the automated chemagic MSM I system according to the manufacturer’s specifications (PerkinElmer Chemagen Technologie GmbH, Baesweiler, Germany). The remaining gDNA samples (*n* = 23) were extracted at external labs and sent to us. Genomic DNA integrity and concentration were evaluated on a Nanodrop instrument (Life Technologies, Carlsbad, CA, USA). Aliquots of gDNA were diluted to 10 ng/μL with ddH_2_O for use in PCRs.

### 4.3. Long-Range PCR primers

Primers for LR PCR were designed on NCBI’s Primer-Blast to be 28 bp (range 26–30) long and to have a melting temperature (Tm) of 67 °C (range 65–68 °C) under default settings [31]. The size of PCR products depended on the target locus and ranged from 4.3 to 19.9 kb (mean size = 15.8 kb, median size = 17.6 kb). Primers for a total of 124 PCRs to amplify the following IRD-associated loci were validated (synthesized by Microsynth AG, Balgach, Switzerland): *PPT1*, *ABCA4*, *CRB1*, *PCARE* (*C2orf71*), *EFEMP1*, *IMPG2*, *PROM1*, *MFSD8*, *CTNNA1*, *GUCA1A*, *GUCA1B*, *PRPH2*, *IMPG1*, *ELOVL4*, *DHS6S1*, *RP1L1*, *RP1*, *CNGB3*, *KCNV2*, *ATOH7*, *PDE6C*, *BEST1*, *C1QTNF5*, *PDE6H*, *RDH5*, *OTX2*, *NR2E3*, *RLBP1*, *GUCY2D*, *FSCN2*, *RAX2*, *TIMP3*, *RS1*, *RPGR*, and *RP2*. The validated primers are listed in Appendix A in the Appendix A.

### 4.4. Long-Range PCR

The LR PCR products used in this study were generated using Takara’s long and accurate (LA) *Taq* polymerases (Takara Bio, Kusatsu, Japan) according to the following PCR mixture: 30 μL total reaction volume, 11.5 μL of ddH_2_O, 50 ng of template gDNA, 3 μL of 10× LA PCR Buffer II (Mg^2+^ plus), 3 μL of 10× Solution S (Solis BioDyne, Tartu, Estonia), 4.8 μL of deoxynucleoside triphosphates (dNTPs) Mixture (2.5 mM each), 1.2 μL of each primer (10 mM), and 0.3 μL of Takara LA *Taq* (5 units/μL). Reactions were performed on a Veriti thermal cycler (Applied Biosystems, Foster City, CA, USA) according to two-step PCR conditions: 94 °C for 2 min, 35 cycles of 98 °C for 10 s and 68 °C for 12 min, followed by 72 °C for 10 min, and hold at 4 °C.

### 4.5. PCR Quality Control and Pooling

The expected size of amplicons was confirmed by electrophoresis on 0.6% agarose gels (run at 60 V) and 1 μL of the reaction mixture was used to measure the amplicon concentration with the Qubit dsDNA High-Sensitivity Assay kit (ThermoFisher, Waltham, MA, USA).

All amplicons that would be sequenced with the same index sequence (either all PCRs of a specific proband or non-overlapping amplicons of different probands) were diluted to 10 ng/μL in 1× Tris-EDTA (TE) buffer (Integrated DNA Technologies, Coralville, IA, USA). The volume of each diluted PCR added to the pool was proportional to its size, for a final pool of at least 130 μL in total volume.

### 4.6. Library Construction and Sequencing

Each pool was sheared on a Covaris M220 (Covaris, Woburn, MA, USA) in a 130 μL Covaris Adaptive Focused Acoustics (AFA) microtube (Covaris, Woburn, MA, USA) to a target size of ~350–400 bp, with the following settings: 50 W peak incident power, 20% duty factor, 200 cycles per burst, 65 s treatment time.

Successful fragmentation was checked by running 1 μL of the sheared pool with a Bioanalyzer High-Sensitivity DNA kit on a Bioanalyzer 2100 instrument (Agilent Technologies, Santa Clara, CA, USA).

The validated pools were then processed with the TruSeq DNA Nano Low-Throughput Library Prep kit and TruSeq DNA Single Index Set A and B, according to the manufacturer’s protocol (Illumina, San Diego, CA, USA).

Molarities of the libraries were calculated and diluted to a 4 nM working concentration. The different libraries that were to be sequenced together (with different indexes) were pooled proportionally to their total genomic target size. Subsequently, these pools were denatured with 0.2N NaOH and loaded into a MiSeq Reagent Kit V2 (300 cycles) cartridge, according to the manufacturer’s recommendations (Illumina, San Diego, CA, USA). Paired-end sequencing (2× 151 cycles) was performed on a MiSeq instrument (Illumina, San Diego, CA, USA).

A detailed step-by-step protocol is provided in the Appendix A (Appendix A).

### 4.7. ABCA4 Capture Sequencing

Custom-capture probes (xGen Lockdown Probe Pools) were designed for the *ABCA4* locus by IDT (Integrated DNA Technologies, Coralville, IA, USA). Genomic DNA libraries were constructed according to the ThruPLEX DNA-seq 96D kit protocol (Takara Bio, Kusatsu, Japan). Libraries were pooled equimolarly for a total of 1600 ng of DNA. Subsequently, the *ABCA4* capture probes were used according to the manufacturer’s protocol to enrich the *ABCA4* locus (Integrated DNA Technologies, Coralville, IA, USA). The resulting libraries were sequenced on a MiSeq according to manufacturer’s recommendations (Illumina, San Diego, CA, USA).

### 4.8. Whole-Exome Sequencing

WES was performed for 152 probands: 23 of them underwent WES in an external facility on an Illumina HiSeq instrument (AtlasBiolabs, Berlin, Germany) or on a SOLiD 5500 xl system (CeGat, Tübingen, Germany), whilst 129 probands were sequenced in-house on a NextSeq 550 (Illumina, San Diego, CA, USA). Of the 129 in-house WES, 31 were captured using the Nextera Rapid Capture Exome kit (Illumina, San Diego, CA, USA), 65 with the TruSeq Exome kit (Illumina, San Diego, CA, USA), and 33 with the xGen Exome Research Panel (Integrated DNA Technologies, Coralville, IA, USA), according to the manufacturers’ protocols.

### 4.9. Exome Spike-In Applications

To assess the feasibility of spike-in to WES experiments to enhance coverage of regions of interest, we tested two alternatives: by adding the custom xGen Lockdown Probe pool for the *ABCA4* locus to the TruSeq Exome kit, capturing probes mix before hybridization, or by adding the processed LR PCR libraries just before final library denaturation.

An aliquot of the xGen Lockdown Probes was diluted to a concentration of 85.5 aM and a total of 250 amol were added to the TruSeq Exome capture mix for spike-in. Conversely, for the LR libraries spike-in, the amount of library to be added was calculated based on the ratio of the total LR library target regions and the WES target region.

### 4.10. Sequencing Data Analysis

Illumina sequencing data was aligned to the human reference genome hg19 with Burrows-Wheeler Aligner (BWA), and variant calling was performed by Genome Analysis Toolkit (GATK) [32,33]. The resulting Variant Call Format files (VCFs) were annotated with Alamut Batch v1.8 (SophiaGenetics, Lausanne, Switzerland) using a gene list corresponding either to the loci present in the RetNet database (*n* = 276) for WES data, or to the target loci for LR PCRs data. CNVs analysis on target-capture sequencing data (*ABCA4* capture and WES results) was performed using panelcn.mops [34] and the SeqNext module of Sequence Pilot v5.2 (JSI medical systems, Ettenheim, Germany).

LR PCR sequencing data of *RPGR*’s exon ORF15 was assembled de novo using SPAdes and the resulting contig was aligned to the reference sequence, as previously described [29,35].

### 4.11. CNV Validation and Breakpoints Characterization

MLPA (MRC Holland, Amsterdam, The Netherlands) was used to validate candidate CNVs identified by panelcn.mops and/or SeqNext in *ABCA4*, *BEST1*, *GUCY2D*, and *RPGRIP1* and to screen for CNVs in *ABCA4*, *BEST1*, and *PRPH2*.

Deletions encompassing exons 20 through 22 of *ABCA4*, exon 2 of *RS1*, exons 2 through 4 of *RP1*, and the entire *KCNV2* gene were identified by panelcn.mops and SeqNext or by comparison of coverage plots. These deletions were verified and characterized by LR PCR. The primers used are listed in Appendix A in the Appendix A. Briefly, primers were designed based on the estimated breakpoints location, LR PCR was performed, and the amplicon was sequenced as described above (see amplicons pool library construction and sequencing). In addition, breakpoints were confirmed by Sanger sequencing (primers in Appendix A in the Appendix A).

To narrow down the breakpoint region of the *KCNV2* deletion, an Infinium CytoSNP-850K BeadChip array analysis was performed according to the manufacturer’s protocol (Illumina, San Diego, CA, USA).

The *ABCA4* deletion was further characterized and refined by TLA (Cergentis, Utrecht, The Netherlands).

### 4.12. Chromosomal Phasing

Segregation analysis was performed if samples from family members were available, either by Sanger sequencing (primers on Appendix A in the Appendix A) or by LR PCR sequencing, as described above.

Alternatively, when no family members were available for testing, read phasing was performed. For this, reads in the Binary Alignment Map (BAM) file were visually inspected for informative heterozygous variants between the two putative compound heterozygous candidate variants (Figure 2).

## Figures and Tables

**Figure 1 ijms-22-01508-f001:**
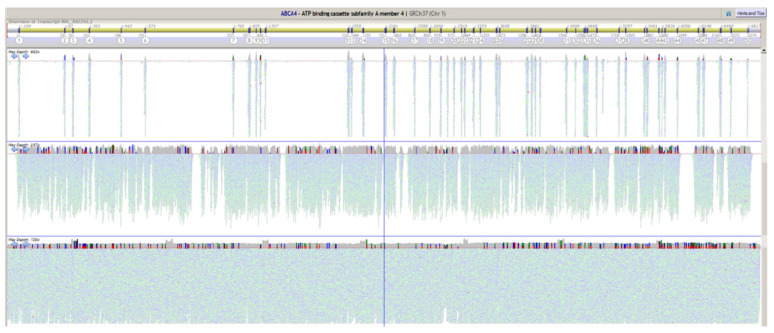
*ABCA4* coverage comparison of different assays. Screenshot of the Alamut Visual software showing the coverage results for the *ABCA4* locus from different next-generation sequencing assays. On the top, the software illustrates the relative exon locations. Below the gene structure, coverage plots for a typical whole-exome sequencing assay (top), custom capture probes assay (middle), and, finally, the long-range polymerase chain reaction method (bottom).

**Figure 2 ijms-22-01508-f002:**
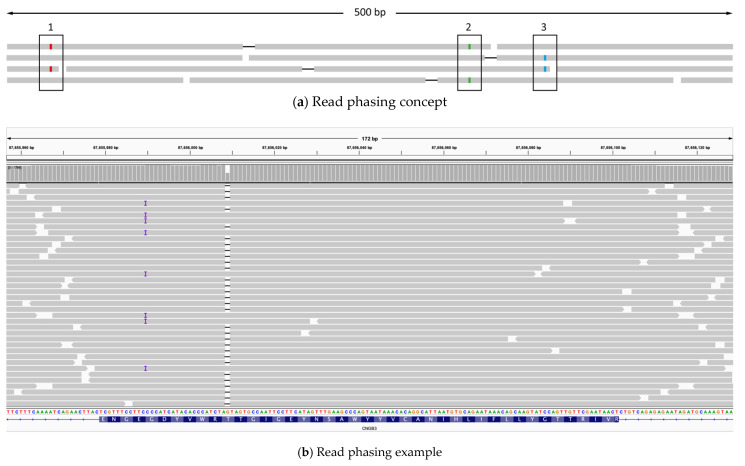
Read phasing: (**a**) Conceptual representation of read phasing. The grey bars represent sequencing reads aligned to the genome. Mate pair reads are connected by a black line. Single nucleotide variants are shown by a colored line and highlighted by black boxes. In this representation, variants at position 1 and 3 are the two putative pathogenic variants. The mate pair read on the top left is evidence that variants 1 and 2 are in cis. On the other hand, mate pair reads on the second and third lines attest that variants 2 and 3 are in trans. Therefore, variants 1 and 3 are in trans. (**b**) Screenshot from Integrative Genomics Viewer (IGV) of a simple example of read phasing from this study. The illustration shows the location of the two potentially pathogenic variants for patient S220 (*CNGB3*:c.1148del and *CNGB3*:c.1167_1168insC).

**Figure 3 ijms-22-01508-f003:**
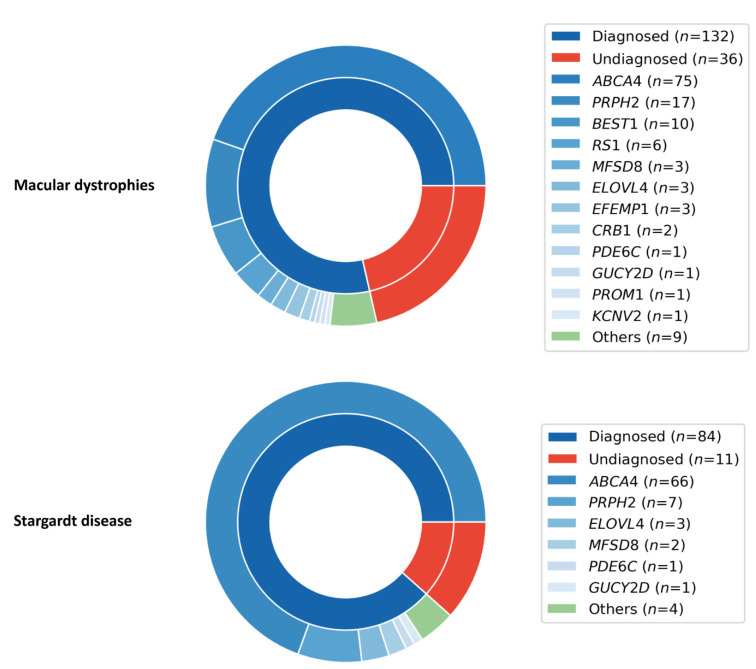
Diagnostic yields and main contributing loci for the macular dystrophies cohort and for the Stargardt disease sub-cohort. Nested pie charts depicting diagnostic yield (inner ring) and the identified contributing loci (outer ring) for macular dystrophies (top) and Stargardt disease (bottom). The loci included in the long-range polymerase chain reaction (PCR) panel are shown in different shades of blue, whilst other retinal diseases-associated loci not included in the panel are shown in green. *ABCA4* and *PRPH2* are the main contributors in both cohorts.

**Table 1 ijms-22-01508-t001:** Established long-range polymerase chain reactions (PCRs) for retinal diseases-associated loci. The first column indicates the names of the loci included in the panel, ordered by chromosome coordinates. The second column highlights the number of PCRs necessary to cover the locus comprehensively, and the last column shows the total size of the genomic target sequence. Abbreviations: bp, base pairs.

Locus	Number of PCRs Needed	Genomic Target Size (bp)
*PPT1*	2	26,746
*ABCA4*	8	137,748
*CRB1*	14	212,898
*PCARE*	1	18,628
*EFEMP1*	5	60,569
*IMPG2*	6	94,768
*PROM1*	7	117,712
*MFSD8*	3	49,747
*CTNNA1*	13	182,694
*GUCA1A*	2	27,191
*GUCA1B*	1	13,548
*PRPH2*	2	32,242
*IMPG1*	11	144,700
*ELOVL4*	2	34,792
*DHS6S1*	1	13,473
*RP1L1*	3	51,877
*RP1*	1	16,930
*CNGB3*	12	172,052
*KCNV2*	1	15,214
*ATOH7*	1	9275
*PDE6C*	3	53,794
*BEST1*	1	17,272
*C1QTNF5*	1	10,536
*PDE6H*	1	11,084
*RDH5*	1	6122
*OTX2*	1	12,758
*NR2E3*	1	11,097
*RLBP1*	1	13,667
*GUCY2D*	1	18,939
*FSCN2*	1	11,574
*RAX2*	1	5845
*TIMP3*	5	67,146
*RS1*	3	34,731
*RPGR*	4	60,808
*RP2*	3	46,521

**Table 2 ijms-22-01508-t002:** Macular dystrophies (MD) cohort summary. Abbreviations: STGD, Stargardt disease; MD, macular dystrophy; BEST, Best disease; ML, malattia leventinese; XLRS, X-linked retinoschisis; SFD, Sorsby fundus dystrophy.

Category	Total	Total Diagnosed	Total Undiagnosed	Mean Age at Referral (Years)	Diagnostic Yield (%)
Overall	168	132	36	37.86	78.6
Stratification group					
Validation	70	70	0	35.60	100.0
No previous testing	85	58	27	40.14	68.2
Missing heritability	13	4	9	35.08	30.8
Clinical diagnosis					
STGD	95	84	11	35.88	88.4
Unspecified MD	45	28	17	40.11	62.2
BEST	11	9	2	35.73	81.8
ML	9	5	4	47.89	55.6
XLRS	6	6	0	32.00	100.0
SFD	1	0	1	63.00	0.0

**Table 3 ijms-22-01508-t003:** Novel likely pathogenic variants. Abbreviations: cNomen, Human Genome Variation Society (HGVS) cDNA-level nucleotide change nomenclature; pNomen, predicted protein-level change nomenclature; gnomAD, genome aggregation database minor allele frequency; ACMG, American College of Medical Genetics and Genomics guidelines; CADD, Combined Annotation-Dependent Depletion.

Locus	cNomen	pNomen	GnomAD Overall (%)	GnomAD Max. (%)	ACMG Class	CADD Score
*ABCA4*	NM_000350.2:c.6731T > A	p.Val2244Glu	0	0	3	27.3
*ABCA4*	NM_000350.2:c.6428T > A	p.Met2143Lys	0	0	4	32.0
*ABCA4*	NM_000350.2:c.6323_6331delinsGGC	p.Met2108_Asn2111delinsArgHis	0	0	4	35
*ABCA4*	NM_000350.2:c.5924G > T	p.Gly1975Val	0	0	4	28.8
*ABCA4*	NM_000350.2:c.5690_5704del	p.Gln1897_Phe1901del	0	0	3	22.2
*ABCA4*	NM_000350.2:c.5691G > T	p.Gln1897His	0	0	3	23.8
*ABCA4*	NM_000350.2:c.5461–6T > C	p.?	0	0	3	14.93
*ABCA4* ^1^	NM_000350.2:c.4958G > A	p.Gly1653Glu	0	0	3	28.1
*ABCA4*	NM_000350.2:c.4609del	p.Thr1537ArgfsTer6	0	0	4	33
*ABCA4*	NM_000350.2:c.4383G > C	p.Trp1461Cys	0.0004	0.003	3	32
*ABCA4*	NM_000350.2:c.3323del	p.Arg1108ProfsTer40	0	0	5	34
*ABCA4*	NM_000350.2:c.3179A > C	p.Gln1060Pro	0.0008	0.0065	4	23.7
*ABCA4* ^2^	NM_000350.2:c.(2918 + 765_2918 + 775)_(3328 + 618_3328 + 662)del	p.Leu973_Asp2273delinsPheMetAlaArgValGluArgSerLeuGlyAsn	0	0	5	
*ABCA4*	NM_000350.2:c.1742C > A	p.Thr581Asn	0	0	3	26.7
*ABCA4*	NM_000350.2:c.1621_1622del	p.Leu541ThrfsTer14	0	0	4	32
*ABCA4*	NM_000350.2:c.727_728dup	p.Tyr245CysfsTer18	0	0	5	26.2
*ABCA4*	NM_000350.2:c.676C > A	p.Arg226Ser	0.0068	0.0163	3	14.13
*CRB1*	NM_201253.2:c.1472A > T	p.Asp491Val	0	0	3	15.86
*CRB1*	NM_201253.2:c.2298G > A	p.Trp766Ter	0	0	5	36
*OPA1*	NM_130837.2:c.2987A > C	p.Lys996Thr	0	0	4	23.8
*PROM1*	NM_006017.2:c.2476G > C	p.Asp826His	0	0	3	32
*MFSD8* ^1^	NM_152778.2:c.670A > T	p.Asn224Tyr	0	0	3	23.9
*GUCA1A*	NM_000409.4:c.333G > C	p.Glu111Asp	0	0	4	22.9
*PRPH2*	NM_000322.4:c.611_626del	p.Tyr204SerfsTer47	0	0	5	33
*PRPH2*	NM_000322.4:c.605G > A	p.Gly202Glu	0	0	4	29.8
*PRPH2*	NM_000322.4:c.512T > G	p.Phe171Cys	0	0	4	27.3
*RP1L1*	NM_178857.5:c.1024_1026delinsCTCCT	p.Arg342LeufsTer22	0	0	4	22
*RP1L1*	NM_178857.5:c.196G > C	p.Asp66His	0	0	3	26.6
*KCNV2* ^3^	NM_133497.3:c.-759_*57289del	p.?	0	0	5	
*KCNV2* ^1^	NM_133497.3:c.1096del	p.Val366TrpfsTer88	0	0	4	13.41
*RGR*	NM_002921.3:c.236G > A	p.Arg79His	0.0032	0.0141	3	37
*BEST1*	NM_004183.3:c.907G > T	p.Asp303Tyr	0	0	5	28.5
*CNGB1*	NM_001297.4:c.2662G > A	p.Ala888Thr	0.0225	0.1145	3	18.28
*CNGB1*	NM_001297.4:c.1658C > A	p.Ala553Glu	0.0004	0.0009	3	21.1
*GUCY2D*	NM_000180.3:c.929C > A	p.Thr310Asn	0	0	3	24.5
*RS1*	NM_000330.3:c.209G > A	p.Gly70Asp	0	0	4	26.7
*RS1*	NM_000330.3:c.150G > A	p.Trp50Ter	0	0	5	35
*RS1* ^4^	NM_000330.3:c.53–717_78 + 262del	p.Ala18_Glu26delinsGluProGlyGlnHisSerLysThrLeu	0	0	5	
*RPGR*	NM_001034853.1:c.2819_2838dup	p.Glu947LysfsTer149	0	0	3	22.4

^1^ the variant has been found in 2 unrelated probands. ^2^ size of the deletion: 3999–4033 bp (NC_000001.10:g.(94507656_94507700)_(94511700_94511710)del). ^3^ size of the deletion: 70,036 bp (NC_000009.11:g.2716981_2787016del). ^4^ size of the deletion: 1005 bp (NC_000023.10:g.18675498_18676502del).

## Data Availability

The data presented in this study are available in the article and the Appendix A. Raw data are not publicly available due to data protection regulations.

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
