# Peer review of "Long-Range PCR-Based NGS Applications to Diagnose Mendelian Retinal Diseases"

_ijms, 2021, doi:10.3390/ijms22041508_

Round 1
Reviewer 1 Report
Very good article, I must say that it is one of the best I have reviewed on the development of new methodologies.
The bibliography is updated, the patient cohort is large, well-structured and allows for pre-application in other laboratories.
There are just a few comments:
- The deletions in RS1 are in hemizygous not homozygous, it is the X chromosome.
Hemizygous (Collins English Dictionary): being or having a gene, esp. one on the X chromosome, that lacks an allelic complement and which therefore always expresses the trait which it carries.
(https://www.collinsdictionary.com/es/diccionario/ingles/hemizygous).
- The MPLA of exons 20 and 22 of ABCA4 in Figure B1 should be repeated.
- In the figures for each of the genes where a deletion has been detected, please put in the heading at what dose the mutation is (e.g. heterozygous).
- The legend of the tables needs to be more clear (e.g. NA?)
Author Response
Very good article, I must say that it is one of the best I have reviewed on the development of new methodologies.
The bibliography is updated, the patient cohort is large, well-structured and allows for pre-application in other laboratories.
Response 1:
We thank reviewer 1 for critically reading our manuscript and the very positive feedback.
There are just a few comments:
Point 1: The deletions in RS1 are in hemizygous not homozygous, it is the X chromosome.
Hemizygous (Collins English Dictionary): being or having a gene, esp. one on the X chromosome, that lacks an allelic complement and which therefore always expresses the trait which it carries.
Point 1: This was corrected on line 182.
Point 2: The MPLA of exons 20 and 22 of ABCA4 in Figure B1 should be repeated.
Point 2: We repeated the experiment. As Coffalyser is not compatible with our informatics infrastructure anymore, we now switched to a different analysis software. Accordingly, we have provided different screenshots, depicting the results of the novel experiment. We adapted the figure legend accordingly.
Point 3: In the figures for each of the genes where a deletion has been detected, please put in the heading at what dose the mutation is (e.g. heterozygous).
Point 3: The genotype information was included in the legends heading for each figure.
Point 4: The legend of the tables needs to be more clear (e.g. NA?)

Point 4: We have added the full description of additional abbreviations used in tables which were not provided in the previous version.
Reviewer 2 Report
This is an original research for molecular diagnosis of inherited retinal diseases with 124 long range-based PCR with primers designed for sequencing 35 genomic loci (including ABCA4, PRPH2, BEST1, CRB1, CNGB3, RP1, and RPGR) that have been known to cause inherited retinal diseases especially macular dystrophy and X-linked retinitis pigmentosa. The methodology used in this study is sophisticated. Authors claimed the method they proposed can be used to do:
(i) as a first-tier assay to reduce genetic testing costs,
(ii) to elucidate missing heritability cases,
(iii) to characterize breakpoints of CNVs at the nucleotide level,
(iv) to extend WES data to non-coding regions by spiking-in long-range PCR libraries, and
(v) to help with phasing of candidate variants.
Comments for this study:
- Line 19, 35 and may other places. Using “RD” to represent retinal diseases is totally confusing. It is believed that RDs represents inherited retinal diseases in this manuscript. Unifying the term in abbreviation for inherited retinal diseases as IRDs is recommended.
- Line 24. CNV is a confusing abbreviation to ophthalmologists. It is better to show the full name “copy number variants” instead of only CNVs when this term appears for the first in this manuscript.
- Line 60. The methodologies for genetic testing evolve rapidly. How do authors define standard genetic testing? Please describe more detail in this place.
- Line 306. Authors have assigned participants to one of three groups: (i) validation cohort (ii) missing heritability cohort (iii) probands without previous genetic testing. What was the basis for dividing them into these three groups? Patients in Group (i) and (ii) may have been tested with the same molecular diagnosis such as WES before. Did Lange-range PCR-based NGS show complete consistence in the causative gene with the previous molecular diagnosis of the cohort A in this study? Please provide more information about previous molecular diagnosis for group (i)
- Of the novel likely pathogenic variants mentioned in Line 215, can these 161 novel casual variants be found within all three groups of the cohort? In Line 463 table A1 shows many patients in cohort (i) with variant 1 and variant 2. How many possible variants in one patient was found? Please provide full names of ACMG and HGMD.
- Line 66 and 70. Lange range PCRs were established for 35 loci associated with “retinal diseases”. It has been shown that maybe more than 300 genes (RETNET; https://sph.uth.edu/retnet/) are associated with inherited retinal diseases. It is curious to know what the criteria for selection of those 35 genes was in the present study. Did author determine the candidate based previous population-based study or according to what kind of a database? It is acquainted that inherited retinal diseases do not have a good consistence between genetic variants and the phenotypes. It may not be easy to determine the sequencing targets based on clinical phenotype and family history when precise clinical assessment is not easy for clinical ophthalmologists (Line 252).
- Line 79. Did the long-range PCR demonstrate the same efficiency of molecular diagnosis for all the 35 loci with a variety of the genomic target length or the PCR number required? Since it is not able to detect deletions spanning primer-binding regions (described in Line 244), more PCR number may be less efficient.
- Line 260. Variants in three genes, ABCA4, PRPH2, and BEST1, explained the 60.7% in the present and 57% of the cases of macular dystrophy in the previously study. However, the detail of families participated in the present study is not described in the manuscript. This retrospective single-center cross-sectional study (reference 11) included only Caucasian patients. Therefore, the finding may only be true for Caucasian in west Europe and hardly represent patients globally all over the world.
- Line 95. LR PCR method indeed provide uniform coverage over the entire locus (Figure S1). However, the depth of sequencing seems vary in part of genes selected such RPGR, PDE6C, CTNNA1, FSCN2 and RP2.
- Line 106 and 237. Authors suggest LR PCR based NGS as a first-tier genetic assay (51.1% diagnosed) to reduce costs for probands without previous genetic testing. The Second-tier WES analysis revealed additional likely pathogenic variants, resulting in a total of 67.0%. It is indicated that the target regions to be sequenced may be personalized according to the clinical phenotype and family history, therefore costs can be reduced compared to standard genetic testing. How the cost of genetic testing can be reduced is not clearly compared in this study except a short mention of 34% at Line 269 within Discussion.
- A large of abbreviations used. Please provide full name when is was used for the first time this manuscript: Line 25 ES, Line 136 MPLA, Line 216 SNVs, Line 226 Table3 ACMG and CADD, Line 245 ADO.
- Line 386. The purpose of Method 4.9 needs to be explained in more detail. Why it is necessary to assess the feasibility of spike-in to WES experiments to enhance coverage of regions of interest is not clarified. Readers may like to know more he working mechanism for the description of “LR libraries can be spiked into an exome library to either enhance coverage of poorly captured exonic regions (such as RPGR’s ORF15) or to obtain coverage of otherwise uncaptured regions.” in Line 287.
Author Response
This is an original research for molecular diagnosis of inherited retinal diseases with 124 long range-based PCR with primers designed for sequencing 35 genomic loci (including ABCA4, PRPH2, BEST1, CRB1, CNGB3, RP1, and RPGR) that have been known to cause inherited retinal diseases especially macular dystrophy and X-linked retinitis pigmentosa. The methodology used in this study is sophisticated. Authors claimed the method they proposed can be used to do:
(i) as a first-tier assay to reduce genetic testing costs,
(ii) to elucidate missing heritability cases,
(iii) to characterize breakpoints of CNVs at the nucleotide level,
(iv) to extend WES data to non-coding regions by spiking-in long-range PCR libraries, and
(v) to help with phasing of candidate variants.
Response 1:
We thank reviewer 1 for critically reading our manuscript and the positive feedback.
Comments for this study:
Point 1: Line 19, 35 and may other places. Using “RD” to represent retinal diseases is totally confusing. It is believed that RDs represents inherited retinal diseases in this manuscript. Unifying the term in abbreviation for inherited retinal diseases as IRDs is recommended.
Point 1: We agree that IRD is the more common abbreviation used when referring to monogenic retinal diseases; we previously opted for ‘retinal diseases’ simply due to word limit constraints in the abstract, and to be consistent with the title, in which we refer to them as Mendelian retinal diseases. We have changed every instance of the abbreviation to the more commonly used “IRD”. This has also required slight changes to the abstract.
Point 2: Line 24. CNV is a confusing abbreviation to ophthalmologists. It is better to show the full name “copy number variants” instead of only CNVs when this term appears for the first in this manuscript.
Point 2: CNV may be confused for choroidal neovascularization, if not otherwise defined. We edited the abstract accordingly.
The CNV abbreviation is also explained in the first instance of the main text (line 66), and other instances remain therefore unchanged.
Point 3: Line 60. The methodologies for genetic testing evolve rapidly. How do authors define standard genetic testing? Please describe more detail in this place.
Point 3: The methodologies for genetic testing have evolved rapidly with the introduction of novel methods. Since 2010-2015 most diagnostic labs working on IRD took advantage of targeted NGS in the form of WES or gene panels (as discussed on lines 45-47). We refer to this strategy as the current “standard” assay for genetic testing in IRDs (we further elucidated this on line 58). More labs may adopt genome sequencing in the future, but this is still likely some time away from adoption as standard practice.
Point 4: Line 306. Authors have assigned participants to one of three groups: (i) validation cohort (ii) missing heritability cohort (iii) probands without previous genetic testing. What was the basis for dividing them into these three groups? Patients in Group (i) and (ii) may have been tested with the same molecular diagnosis such as WES before. Did long-range PCR-based NGS show complete consistence in the causative gene with the previous molecular diagnosis of the cohort A in this study? Please provide more information about previous molecular diagnosis for group (i)
Point 4: With regard to the first question: patients in group (i) and (ii) had been previously tested with WES (we added this information on section 4.1). The difference between the two groups is that WES allowed for the identification of a molecular diagnosis for patients in group (i), whilst patients in group (ii) remained without a molecular diagnosis after WES. Also, patients in group (ii) were selected because a likely pathogenic variant was identified in a recessively inherited gene of the LR PCR panel.
Regarding the second question: patients of group (i) were selected based on their previous molecular diagnosis and used for validation purposes. The long-range PCR method showed 100% sensitivity to all previously identified likely pathogenic variants, and confirmed previous WES results.
Point 5: Of the novel likely pathogenic variants mentioned in Line 215, can these 161 novel casual variants be found within all three groups of the cohort? In Line 463 table A1 shows many patients in cohort (i) with variant 1 and variant 2. How many possible variants in one patient was found? Please provide full names of ACMG and HGMD.
Point 5: The 39 novel likely pathogenic variants are listed in Table 3. We found a total of 161 unique likely pathogenic variants that we suspect are disease-causing in the three cohorts. Many of these variants (122/161) have been previously described as being pathogenic, whilst 39 have never been associated with disease, to our knowledge. The first report of the corresponding variant is reported in the “Ref.” column of Tables A1, A2, and A3.
Tables A1, A2, and A3 only report the variants that were selected as disease-relevant in the corresponding patient. If a second variant is listed in the table (Variant 2), the variants are inherited in a recessive manner. It does not correlate with the number of variants identified in the patient.
We provided full name for ACMG and HGMD in the Tables captions.
Point 6: Line 66 and 70. Long-range PCRs were established for 35 loci associated with “retinal diseases”. It has been shown that maybe more than 300 genes (RETNET; https://sph.uth.edu/retnet/) are associated with inherited retinal diseases. It is curious to know what the criteria for selection of those 35 genes was in the present study. Did author determine the candidate based previous population-based study or according to what kind of a database? It is acquainted that inherited retinal diseases do not have a good consistence between genetic variants and the phenotypes. It may not be easy to determine the sequencing targets based on clinical phenotype and family history when precise clinical assessment is not easy for clinical ophthalmologists (Line 252).
Point 6: The first criterion was size (not larger than 250 kb). Then, we included all loci that have been associated with a form of macular dystrophy or retinal disease with macular involvement:
[BEST1, C1QTNF5, CNGB3, EFEMP1, ELOVL4, FSCN2, GUCA1B, PROM1, PRPH2, TIMP3, ABCA4, RPGR, RS1, DHS6S1 (PRDM13)]1, CRB12, MFSD83, [IMPG1, IMPG2, CTNNA1]4, PPT15, RP1L16, OTX27, RDH58, GUCA1A9, GUCY2D10, RAX211
or with X-linked RP:
[RPGR, RP2]12
The other RP and COD/CRD loci were selected based on necessity to tackle missing heritability in specific patients (RP1, PCARE, KCNV2, PDE6C, PDE6H, RDH5, NR2E3).
We acknowledge the challenges in formulating a specific clinical diagnosis. Clinical variability complicates this task. Genetic heterogeneity is extensive and, when combined with the challenges of establishing a precise clinical diagnosis, selecting a relatively small number of candidate genes may seem like an unreasonable task. However, it has been reported by several studies that relatively few genes contribute to a large portion of IRD patients, in particular ABCA4. We have added a sentence on line 68-69 to point this out.
For these reasons, we proposed the method to have different applications, based on the specific task on hand.
Since a few genes contribute to a large portion of patients, sequencing these loci as a first-tier assay (e.g. ABCA4, PRPH2, and BEST1 for non-RP, and RPGR and RP2 for X-linked RP) would decrease costs and avoid incidental findings (application (i)).
On the other hand, if a patient remains undiagnosed after WES, with only one likely pathogenic variant in a recessively inherited, sequencing that specific locus comprehensively could facilitate identification of a second likely pathogenic variant that was previously undetected by WES (application (ii)).
Point 7: Line 79. Did the long-range PCR demonstrate the same efficiency of molecular diagnosis for all the 35 loci with a variety of the genomic target length or the PCR number required? Since it is not able to detect deletions spanning primer-binding regions (described in Line 244), more PCR number may be less efficient.
Point 7: Number of PCR required to cover a specific locus did not correlate with the efficiency to detect variants. However, as pointed out in the manuscript, the method has limitations when it comes to detecting copy number variants and structural variants, when they are larger than the PCR fragment over the affected region and/or are located in overlapping PCR regions. We tried to reduce this risk by designing the amplicons to have 1-2kb overlapping regions, when more than one PCR was needed to cover the locus.
Nevertheless, we showed the method to be highly sensitive for the detection of CNVs within a single PCR fragment. Moreover, as these fragments are large, the method is sensitive to CNVs that are often missed by WES CNV analysis and array-based CNV analysis.
Point 8: Line 260. Variants in three genes, ABCA4, PRPH2, and BEST1, explained the 60.7% in the present and 57% of the cases of macular dystrophy in the previously study. However, the detail of families participated in the present study is not described in the manuscript. This retrospective single-center cross-sectional study (reference 11) included only Caucasian patients. Therefore, the finding may only be true for Caucasian in west Europe and hardly represent patients globally all over the world.
Point 8: We agree with the Reviewer. Birtel et al. 2017 describes a cohort of MD, COD, and CRD patients of Caucasian origin. We do not provide details on ethnicity, as this information is often not provided to us by referring physicians. We can, however, say that the majority of patients included in our study is also of Caucasian origin. Last week, a report based on a very large cohort in Spain was published13 (this new publication has been cited in the revised manuscript). In this cohort, ABCA4, USH2A, RS1, CRB1, and RHO were the most frequently mutated genes in the diagnosed patients. When considering only non-RP cases (MD, COD, CRD), ABCA4, PRPH2, RS1, and BEST1 were again reported to be the most prominent cause of disease. Another large study also described ABCA4 as being the most commonly mutated gene in a large (1000 patients) IRD cohort14.
We are not aware of other large studies that report diagnostic yield and specific findings in other ethnicities. A relatively small Korean study reported genetic findings for 86 IRD patients; within the diagnosed patients, the most common locus was again ABCA4.15
We were unable to identify any articles reporting genetic findings in other generalized IRD or MD cohorts of non-Caucasian ethnicities. However, one study (Jiang et al. 2016) reported ABCA4 sequencing findings in a Chinese cohort composed of STGD and CRD patients. This study found that likely pathogenic variants in ABCA4 explained disease in 63% of the cohort, and that 10% of their cohort carried a single likely pathogenic variant in ABCA4. Of course, this cohort is not representative of all IRD patients, as it is enriched with STGD patients.
These findings, however, nicely illustrate the main messages of the present study: (i) sequencing of the “usual suspects” based on clinical diagnosis and family history can reduce testing-related costs and incidental findings, and (ii) screening for “second hits” in intronic regions in patients undiagnosed after WES can increase diagnostic yield.
Point 9: Line 95. LR PCR method indeed provide uniform coverage over the entire locus (Figure S1). However, the depth of sequencing seems vary in part of genes selected such RPGR, PDE6C, CTNNA1, FSCN2 and RP2.
Point 9: That is correct. Depth of sequencing can be variable depending mainly on PCR efficiency, which may be optimized by designing novel primers, if required. We do not consider it a problem, as coverage is largely exceeding minimal requirements.
Point 10: Line 106 and 237. Authors suggest LR PCR based NGS as a first-tier genetic assay (51.1% diagnosed) to reduce costs for probands without previous genetic testing. The Second-tier WES analysis revealed additional likely pathogenic variants, resulting in a total of 67.0%. It is indicated that the target regions to be sequenced may be personalized according to the clinical phenotype and family history, therefore costs can be reduced compared to standard genetic testing. How the cost of genetic testing can be reduced is not clearly compared in this study except a short mention of 34% at Line 269 within Discussion.
Point 10: The figure provided in the discussion (34% of material costs saved) was calculated based on the macular dystrophy cohort presented. We calculated the material costs for sequencing ABCA4/PRPH2/BEST1 (top 3 contributors) using the LR PCR method to be around 80 CHF/sample. WES, on the other hand, costs around 300 CHF/sample, according to our calculations. The proposed tiered strategy would have cost 33’240 CHF for the cohort (n=168). Direct WES would have costed 50’400.
Calculation:
Tiered-strategy:
168 patients x 80 CHF/patient (rough material costs for LR-based sequencing of top 3 loci) = 13’440 CHF+(168 patients – 102 diagnosed patients) x 300 CHF/patient (rough material costs for WES) =19’800 CHF=33’240 CHF
WES strategy:
168 patients x 300 CHF/patient = 50’400 CHF
Difference
50’400 CHF – 33’240 CHF = 17’160 CHF
Percentage
17’160/50’400*100 = 34%
This calculation does not include personnel and analysis costs, which would probably further increase the savings gained by the tiered strategy. We did not provide further details as these costs can vary widely based on many factors, including but not limited to sequencing instrument, country-specific material costs, personnel salaries, and cohort composition.
Similarly, Stone et al. 2017 report their tiered-testing strategy to have reduced costs of genetic testing for their large IRD cohort14.
Point 11: A large of abbreviations used. Please provide full name when is was used for the first time this manuscript: Line 25 ES, Line 136 MPLA, Line 216 SNVs, Line 226 Table3 ACMG and CADD, Line 245 ADO.
Point 11: We thank the Reviewer for drawing our attention to these omissions. ES was a typo (meant to be WES). ADO is defined on line 120. The other abbreviations were corrected at the appropriate part of the manuscript. An “Abbreviations” section was added to each Table caption.
Point 12: Line 386. The purpose of Method 4.9 needs to be explained in more detail. Why it is necessary to assess the feasibility of spike-in to WES experiments to enhance coverage of regions of interest is not clarified. Readers may like to know more he working mechanism for the description of “LR libraries can be spiked into an exome library to either enhance coverage of poorly captured exonic regions (such as RPGR’s ORF15) or to obtain coverage of otherwise uncaptured regions.” in Line 287.
Point 12: The rationale behind this concept is centred around the fact that we had previously experienced coding regions of interest not being covered during WES experiments (such as RPGR’s ORF15 and PCARE’s exon 1). Additionally, in the manuscript we discuss the fact that there is increasing evidence for a substantial contribution of deep intronic regions in disease, as shown in the example of ABCA4 deep-intronic variants in STGD1 patients. Intronic regions should therefore ideally be included in genetic testing for loci with published deep-intronic variants, such as ABCA4.
To our knowledge, this approach was never previously tested, and it therefore needed a proof-of-concept study. We hypothesised that by adding LR PCR libraries into the WES captured library it would be possible to acquire data on both deep-intronic regions (not covered by WES) and poorly captured exons (e.g. ORF15 and PCARE exon 1) in a single experiment. It is similar to combining the first application presented (tiered testing strategy) into a single step. This approach has the advantage of reducing the time and financial resources needed to perform the two assays.
We discuss the feasibility of the application and point out limitations and challenges in diluting the spike-in libraries correctly. The protocol to produce the LR PCR libraries is described in detail in the supplementary materials. The WES libraries are produced according to the manufacturer’s protocol. To perform spike-in, the LR PCR libraries need to be diluted based on the ratio of the total target region size included in the library and the total WES target region size, as explained on lines 440-442.
- Berger, W., Kloeckener-Gruissem, B. & Neidhardt, J. The molecular basis of human retinal and vitreoretinal diseases. Prog. Retin. Eye Res. 29, 335–375 (2010).
- Tsang, S. H. et al. Whole exome sequencing identifies CRB1 defect in an unusual maculopathy phenotype. Ophthalmology (2014). doi:10.1016/j.ophtha.2014.03.010
- Roosing, S. et al. Mutations in MFSD8, encoding a lysosomal membrane protein, are associated with nonsyndromic autosomal recessive macular dystrophy. Ophthalmology 122, 170–179 (2015).
- Rahman, N., Georgiou, M., Khan, K. N. & Michaelides, M. Macular dystrophies: Clinical and imaging features, molecular genetics and therapeutic options. Br. J. Ophthalmol. 104, 451–460 (2020).
- Metelitsina, T. I., Waggoner, D. J. & Grassi, M. A. Batten Disease Caused by A Novel Mutation in the PPT1 Gene. Retin. Cases Br. Reports 10, 211–213 (2016).
- Akahori, M. et al. Dominant mutations in RP1L1 are responsible for occult macular dystrophy. Am. J. Hum. Genet. 87, 424–429 (2010).
- Vincent, A. et al. OTX2 mutations cause autosomal dominant pattern dystrophy of the retinal pigment epithelium. J. Med. Genet. 51, 797–805 (2014).
- Nakamura, M., Skalet, J. & Miyake, Y. RDH5 gene mutations and electroretinogram in fundus albipunctatus with or without macular dystrophy: RDH5 mutations and ERG in fundus albipunctatus. Doc. Ophthalmol. 107, 3–11 (2003).
- Chen, X. et al. GUCA1A mutation causes maculopathy in a five-generation family with a wide spectrum of severity. Genet. Med. 19, 945–954 (2017).
- Downes, S. M. et al. Autosomal dominant cone-rod dystrophy with mutations in the guanylate cyclase 2D gene encoding retinal guanylate cyclase-1. Arch. Ophthalmol. 119, 1667–1673 (2001).
- Yang, P., Chiang, P. W., Weleber, R. G. & Pennesi, M. E. Autosomal dominant retinal dystrophy with electronegative waveform associated with a novel RAX2 mutation. JAMA Ophthalmol. 133, 653–661 (2015).
- Sharon, D. et al. RP2 and RPGR Mutations and Clinical Correlations in Patients with X-Linked Retinitis Pigmentosa. Am. J. Hum. Genet. 73, 1131–1146 (2003).
- Perea-Romero, I. et al. Genetic landscape of 6089 inherited retinal dystrophies affected cases in Spain and their therapeutic and extended epidemiological implications. Sci. Rep. 11, 1526 (2021).
- Stone, E. M. et al. Clinically Focused Molecular Investigation of 1000 Consecutive Families with Inherited Retinal Disease. Ophthalmology 124, 1314–1331 (2017).
- Kim, M. S. et al. Genetic mutation profiles in Korean patients with inherited retinal diseases. J. Korean Med. Sci. 34, (2019).
Reviewer 3 Report
Determination of genomic variants reveals the molecular basis of heritable disorders. It also aids the prediction of the transmission of genetic disorders often allowing preventive measures to limit the severity of a disease.
In the present manuscript, the Authors describe NGS based protocol for genetic diagnosis of heritable retinal diseases. The technique is based on a long-range PCR library preparation covering preselected loci that are frequently mutated in retinal disorders. The preselection of loci allows achieving good sequencing coverage that translates to high-quality sequence data with little assay cost. On the group of 108 samples from patients with a known genetic background, the Authors demonstrate that their method is sensitive (100% detection). Authors also show that the technique can detect new variants that are likely associated with the disease if such variants occur within selected loci. The corresponding author is an expert in the field. The manuscript is clearly written and of good quality. However, some issues should be addressed before publication.
The data is presented on the figures and tables, which partially lack clarity. There are three different types of table numbers (example: Table 1, Table A1, Table S1), which I find confounding. Perhaps it would be good to use only one type of supplementary material? Some figures, presenting screenshots from alignment visualization software, contain numerous small details that are not readable at a given size but increase figure complexity (example Fig.1)
A clear comparison of the proposed workflow to accepted diagnostic protocols would be very helpful. This could be presented as a table that would summarise time, workload, cost of the test, the amount of starting material needed, and the resolution at which genomic changes are detected. This would clarify what the reference for declared cost efectivnes of the proposed protocol was. Also, isn’t the need to prepare ~150 PCR reactions per sample to cover 35 loci making the protocol labor intensive? Is this included in the cost calculation? Is this step automated?
The approach is based on the assumption that the disease most likely results from the genetic changes in the regions (loci) that were previously associated with such disorders. The 35 investigated loci were selected out of ~300 loci associated with retinal diseases. The selection rationale should be clearly explained.
In paragraph 2.6 “Macular dystrophies cohort” (also Table 2 and Fig. 3), diagnostic yields of 78.6% overall and 88.4% for STGD are highlighted. As the analyzed samples include the validation group, which was known to contain mutations within 35 amplified loci, presented numbers are artificially high and do not reflect the testing protocol performance.
Minor
- ‘CNV’ is not spelled out in the abstract
- What is a possible reason for the lack of the amplification observed for several introns?
- Figure 3 is cut (i.e., it extends beyond the printout page)
- Were the newly discovered variants included in the public database?
- I found it confusing if all tested subjects were analyzed with the same set of 35 loci or the set was further limited in some cases? This could be clarified in the text.
- MLPA is not spelled out
Author Response
Determination of genomic variants reveals the molecular basis of heritable disorders. It also aids the prediction of the transmission of genetic disorders often allowing preventive measures to limit the severity of a disease.
In the present manuscript, the Authors describe NGS based protocol for genetic diagnosis of heritable retinal diseases. The technique is based on a long-range PCR library preparation covering preselected loci that are frequently mutated in retinal disorders. The preselection of loci allows achieving good sequencing coverage that translates to high-quality sequence data with little assay cost. On the group of 108 samples from patients with a known genetic background, the Authors demonstrate that their method is sensitive (100% detection). Authors also show that the technique can detect new variants that are likely associated with the disease if such variants occur within selected loci. The corresponding author is an expert in the field. The manuscript is clearly written and of good quality. However, some issues should be addressed before publication.
Response 1:
We thank the Reviewer for critically reading our manuscript and the positive feedback.
Point 1: The data is presented on the figures and tables, which partially lack clarity. There are three different types of table numbers (example: Table 1, Table A1, Table S1), which I find confounding. Perhaps it would be good to use only one type of supplementary material?
Point 1: We agree that the organization of the materials could be considered confusing. The article type allows for the inclusion of “Appendix” material that is part of the main article file (Table A1 nomenclature). We believe that including the genetic findings tables in the main text file is vital, as they report a large part of the results discussed. However, they are likely too large to be included in the main structure of the article (in particular Table A1, 4 pages long), as they cover 6 entire pages in total.
The same is true for Figures B1-B3: we feel it is important to include them in the main article document, but they may be too disrupting if inserted in the main structure of the article, as they cover 3 pages.
We added some navigation information in the article when referring to either Appendix or Supplementary materials.
Point 2: Some figures, presenting screenshots from alignment visualization software, contain numerous small details that are not readable at a given size but increase figure complexity (example Fig.1)
Point 2: We acknowledge that the alignment figures (Figure 1 and Figure 2b) have details that can be difficult to read on a A4 file. However, we believe that the visible details are enough to convey our message. In addition, the figures will be available online at high resolution, enabling readers to increase the screen size of relevant text.
Point 3: A clear comparison of the proposed workflow to accepted diagnostic protocols would be very helpful. This could be presented as a table that would summarise time, workload, cost of the test, the amount of starting material needed, and the resolution at which genomic changes are detected. This would clarify what the reference for declared cost effectiveness of the proposed protocol was. Also, isn’t the need to prepare ~150 PCR reactions per sample to cover 35 loci making the protocol labor intensive? Is this included in the cost calculation? Is this step automated?
Point 3: Unfortunately, it is not feasible make such a systematic comparison as there are simply too many factors and variables to be displayed on a legible table. We do not suggest the use of the 124 PCRs as a gene panel to sequence for each IRD/MD patient. Instead, we provide validated primers for 35 loci that can be sequenced based on clinical diagnosis.
For example, the first application presented (tiered genetic testing strategy) is based on the use of the technique for as-yet untested patients for some of the most likely causative genes tailored to the specific clinical phenotype: e.g. ABCA4/PRPH2/BEST1 for unspecified MD, ABCA4/PRPH2/ELOVL4 for STGD, and RPGR/RP2 for X-linked RP.
The figure provided in the discussion (34% of material costs saved) was calculated based on the macular dystrophy cohort presented. We calculated the material costs for sequencing ABCA4/PRPH2/BEST1 (top 3 contributors) using the LR PCR method to be around 80 CHF/sample. WES, on the other hand, costs around 300 CHF/sample, according to our calculations. The proposed tiered strategy would have cost 33’240 CHF for the cohort (n=168). Direct WES would have costed 50’400.
Calculation:
Tiered-strategy:
168 patients x 80 CHF/patient (rough material costs for LR-based sequencing of top 3 loci) = 13’440 CHF+(168 patients – 102 diagnosed patients) x 300 CHF/patient (rough material costs for WES) =19’800 CHF=33’240 CHF
WES strategy:
168 patients x 300 CHF/patient = 50’400 CHF
Difference
50’400 CHF – 33’240 CHF = 17’160 CHF
Percentage
17’160/50’400*100 = 34%
This calculation does not include personnel and analysis costs, which would probably further increase the savings. We did not provide further details as this figure can vary widely based on many factors, including but not limited to sequencing instrument, country-specific material costs, personnel salaries, and cohort composition, how many samples are pooled for each sequencing run, etc.
Similarly, Stone et al. 2017 report their tiered-testing strategy to have reduced costs of genetic testing for their large IRD cohort1.
Point 4: The approach is based on the assumption that the disease most likely results from the genetic changes in the regions (loci) that were previously associated with such disorders. The 35 investigated loci were selected out of ~300 loci associated with retinal diseases. The selection rationale should be clearly explained.
Point 4: The first criterion was size (not larger than 250 kb). Then, we included all loci that have been associated with a form of macular dystrophy or retinal disease with macular involvement:
[BEST1, C1QTNF5, CNGB3, EFEMP1, ELOVL4, FSCN2, GUCA1B, PROM1, PRPH2, TIMP3, ABCA4, RPGR, RS1, DHS6S1 (PRDM13)]2, CRB13, MFSD84, [IMPG1, IMPG2, CTNNA1]5, PPT16, RP1L17, OTX28, RDH59, GUCA1A10, GUCY2D11, RAX212
or with X-linked RP:
[RPGR, RP2]13
The other RP and COD/CRD loci were selected based on necessity to tackle missing heritability in specific patients (RP1, PCARE, KCNV2, PDE6C, PDE6H, RDH5, NR2E3).
Point 5: In paragraph 2.6 “Macular dystrophies cohort” (also Table 2 and Fig. 3), diagnostic yields of 78.6% overall and 88.4% for STGD are highlighted. As the analyzed samples include the validation group, which was known to contain mutations within 35 amplified loci, presented numbers are artificially high and do not reflect the testing protocol performance.
Point 5: We did indeed include samples from the validation cohort; however, we included all macular dystrophy patients received from our center. It does therefore represent the diagnostic yield for macular dystrophy for our center. The study included all previously diagnosed MD patients (part of the validation cohort), all untested MD patients, and MD patients unsolved by WES (part of missing heritability cohort).
Point 6:
Minor
- ‘CNV’ is not spelled out in the abstract
- Thanks for pointing this out, we have modified the abstract accordingly
- What is a possible reason for the lack of the amplification observed for several introns?
- We could not pinpoint any specific reason as to why certain intronic region could not be amplified. It has been reported in the literature that high GC-contents often inhibit PCR amplification. However, the intronic regions we could not amplify are characterized by low GC-content (range 25.3-40.8%, mean 33.2%). We also noticed that all of those regions contained A- and T-rich simple (microsatellite) and low complexity repeats, which partially explains why GC-content is low. We cannot, however, formulate a hypothesis as to how AT-rich sequences would influence successful amplification.
- Even though GC-content is relatively low, we cannot exclude that these regions form secondary structures inhibiting amplification.
- Finally, another possible explanation is that these regions may be hotspots for nicking of DNA.
- We could include these observations as part of the discussion, if deemed helpful.
- Figure 3 is cut (i.e., it extends beyond the printout page)
- This has been corrected
- Were the newly discovered variants included in the public database?
- Not yet, but all will be submitted to ClinVar and Varsome when the manuscript is accepted for publication, and ABCA4/RPGR variants will be submitted to LOVD
- I found it confusing if all tested subjects were analyzed with the same set of 35 loci or the set was further limited in some cases? This could be clarified in the text.
- No, none of the subjects were tested for all loci with the LR PCR technique. Only a subset of loci was selected for first-tier sequencing, based on clinical diagnosis and family history. Second-tier LR PCR for the missing heritability cohort focused on recessive loci where 1 likely pathogenic variant had been identified by WES.
- We added clarifications on this regard on lines 123, 138, and 294-295
- MLPA is not spelled out
- This has been corrected
- Stone, E. M. et al. Clinically Focused Molecular Investigation of 1000 Consecutive Families with Inherited Retinal Disease. Ophthalmology 124, 1314–1331 (2017).
- Berger, W., Kloeckener-Gruissem, B. & Neidhardt, J. The molecular basis of human retinal and vitreoretinal diseases. Prog. Retin. Eye Res. 29, 335–375 (2010).
- Tsang, S. H. et al. Whole exome sequencing identifies CRB1 defect in an unusual maculopathy phenotype. Ophthalmology (2014). doi:10.1016/j.ophtha.2014.03.010
- Roosing, S. et al. Mutations in MFSD8, encoding a lysosomal membrane protein, are associated with nonsyndromic autosomal recessive macular dystrophy. Ophthalmology 122, 170–179 (2015).
- Rahman, N., Georgiou, M., Khan, K. N. & Michaelides, M. Macular dystrophies: Clinical and imaging features, molecular genetics and therapeutic options. Br. J. Ophthalmol. 104, 451–460 (2020).
- Metelitsina, T. I., Waggoner, D. J. & Grassi, M. A. Batten Disease Caused by A Novel Mutation in the PPT1 Gene. Retin. Cases Br. Reports 10, 211–213 (2016).
- Akahori, M. et al. Dominant mutations in RP1L1 are responsible for occult macular dystrophy. Am. J. Hum. Genet. 87, 424–429 (2010).
- Vincent, A. et al. OTX2 mutations cause autosomal dominant pattern dystrophy of the retinal pigment epithelium. J. Med. Genet. 51, 797–805 (2014).
- Nakamura, M., Skalet, J. & Miyake, Y. RDH5 gene mutations and electroretinogram in fundus albipunctatus with or without macular dystrophy: RDH5 mutations and ERG in fundus albipunctatus. Doc. Ophthalmol. 107, 3–11 (2003).
- Chen, X. et al. GUCA1A mutation causes maculopathy in a five-generation family with a wide spectrum of severity. Genet. Med. 19, 945–954 (2017).
- Downes, S. M. et al. Autosomal dominant cone-rod dystrophy with mutations in the guanylate cyclase 2D gene encoding retinal guanylate cyclase-1. Arch. Ophthalmol. 119, 1667–1673 (2001).
- Yang, P., Chiang, P. W., Weleber, R. G. & Pennesi, M. E. Autosomal dominant retinal dystrophy with electronegative waveform associated with a novel RAX2 mutation. JAMA Ophthalmol. 133, 653–661 (2015).
- Sharon, D. et al. RP2 and RPGR Mutations and Clinical Correlations in Patients with X-Linked Retinitis Pigmentosa. Am. J. Hum. Genet. 73, 1131–1146 (2003).